# Error Prediction of Air Quality at Monitoring Stations Using Random Forest in a Total Error Framework

**DOI:** 10.3390/s21062160

**Published:** 2021-03-19

**Authors:** Jean-Marie Lepioufle, Leif Marsteen, Mona Johnsrud

**Affiliations:** NILU—Norwegian Institute for Air Research, Postboks 100, 2027 Kjeller, Norway; lm@nilu.no (L.M.); mj@nilu.no (M.J.)

**Keywords:** air quality, quality control, Random Forest, error prediction, total error framework

## Abstract

Instead of a flag valid/non-valid usually proposed in the quality control (QC) processes of air quality (AQ), we proposed a method that predicts the *p*-value of each observation as a value between 0 and 1. We based our error predictions on three approaches: the one proposed by the Working Group on Guidance for the Demonstration of Equivalence (European Commission (2010)), the one proposed by Wager (Journal of MachineLearningResearch, 15, 1625–1651 (2014)) and the one proposed by Lu (Journal of MachineLearningResearch, 22, 1–41 (2021)). Total Error framework enables to differentiate the different errors: input, output, structural modeling and remnant. We thus theoretically described a one-site AQ prediction based on a multi-site network using Random Forest for regression in a Total Error framework. We demonstrated the methodology with a dataset of hourly nitrogen dioxide measured by a network of monitoring stations located in Oslo, Norway and implemented the error predictions for the three approaches. The results indicate that a simple one-site AQ prediction based on a multi-site network using Random Forest for regression provides moderate metrics for fixed stations. According to the diagnostic based on predictive qq-plot and among the three approaches used in this study, the approach proposed by Lu provides better error predictions. Furthermore, ensuring a high precision of the error prediction requires efforts on getting accurate input, output and prediction model and limiting our lack of knowledge about the “true” AQ phenomena. We put effort in quantifying each type of error involved in the error prediction to assess the error prediction model and further improving it in terms of performance and precision.

## 1. Introduction

Air quality (AQ) monitoring is of great importance for measuring air quality in urban areas and establish air quality control strategies. To comply with environmental directives from the European Commission, a network of highly accurate air quality stations is deployed in cities. In addition to being used by policy makers, and people, AQ observations are extensively used by scientists. For instance, researchers use AQ monitoring networks to evaluate low-cost sensors [1], to correct low-cost sensors [2], to assess urban dispersion models [3], to predict AQ Index [4], to produce data fusion [5,6], to implement data assimilation and to validate it [7], among others. Assessment of ambient AQ monitoring is regulated by the European Directive 2008/50/EC “on ambient air quality and cleaner air for Europe” [8]. The directive requires the use of reference methods when measuring air quality ([8], Annex VI). In addition, specific Data Quality Objectives (DQO) for the quality of the reported data must be respected ([8], Annex I). In term of AQ monitoring, Norwegian cities are responsible for:(i)Buying fixed air quality monitoring stations that respect DQOs.(ii)Field operation and quality control according to procedures following standards developed by the European Committee for Standardization (CEN).

The Norwegian Institute for Air Research (NILU), acting as the National reference laboratory for air, is responsible for:(i)Acquiring the measurements from the data logger into an environmental database hosted on its servers.(ii)Helping cities respecting protocols for the maintenance and the data quality control.(iii)Checking the entire Norwegian AQ observation and sending quarterly reports to the Norwegian Environmental Agency as part of preparing for the national AQ reporting to the European Commission.

Not interested in AQI, the Air Quality Index (AQI) is one of the most commonly used indexes at present. This index considers the concentration of six pollutants (CO, NO2, O3, SO2, PM10 and PM2.5).

### 1.1. Outliers and Their Detection Methods

As a result of instrument malfunctions, specific weather events, harsh environments or human mistakes, AQ measurements can be contaminated by outliers. Outliers do not have a formal definition, but many authors define them as measurements not consistent with most of the dataset. Their presence has a masking effect if the outliers are considered as normal measurements and thus are not removed from the dataset. On the other hand, if a measurement is wrongly labeled as an outlier, then it represents a swamping effect.

Patterns of common outliers visible in univariate timeseries are well known [9]: (i) spikes are short-duration (often single samples) positive or negative peaks in data, (ii) a noise anomaly manifests itself as a sudden, unexpected, increase in measurement variance, (iii) low variance in the signal, (iv) no change in the signal shows a constant anomaly, (v) drift, expressed by an offset between the measured values and the original signal. Unfortunately, these patterns are hidden into complex physical phenomena. Indeed, pollutants show particularly rich patterns of variations in space and time on multiple scales. These variations are driven by complex processes of chemical reactions, atmospheric transport, emissions, and depositions. Outlier detection has been extensively studied and applied in many fields such as meteorology, chemical engineering, environmental monitoring, and smart buildings energy consumption, among others. The amount of outlier detection methods and their diversity is wide. They are based either on uni-variate or multi-variate timeseries, and cover unsupervised classification of observations [10,11], supervised classification of observation with label [12], unsupervised regression with threshold [13], supervised regression with threshold [14,15].

### 1.2. Importance of Weighting the Quality of an Observation

AQ experts responsible for quality control (QC) use their knowledge expertise in metrology, chemistry, meteorology, empirical knowledge about each AQ station, its neighborhood environment and the everyday event information (traffic jam, traffic accident, road works, etc.) in order to label AQ observations as valid or as non-valid. However, QC experts might have difficulties affirming a value is 100% valid or 100% non-valid. Moreover, a value that is not completely correct might bring some relevant information. Scientists are used to working with weighted data into their scientific pipeline, for instance to evaluate the importance of observations for a data fusion purpose [5].

In this paper, the weight characterizing an observation corresponds to the *p*-value of its value given an ensemble of quantiles determined by an empiric or a theoretic expression. In this case, we do not aim at providing the best deterministic prediction, but the best predictive distribution.

### 1.3. AQ Prediction and Its Predictive Uncertainty

For the purpose of quality control, AQ prediction with proper uncertainty quantification is crucial. Indeed, its outcome will be used for weighting the quality of observations. Our study does not aim at providing the best deterministic prediction, but instead at providing the best predictive uncertainty or in other words, estimating the error prediction.

**Deep neural networks (DNN)** have achieved high performance in multi-timeseries AQ prediction. Among others, one can cite methods based on auto-encoder [16], on LSTM [17], on LSTM-FC [18], on CNN-LSTM [19] and on graph-CNN-LSTM [20]. Furthermore, methods such as Bayesian DNNs [21,22,23,24] and deep ensemble [25] enable quantifying predictive uncertainty. Having in mind the adaptation of AQ quality control onto edge solution with embedded sensor nodes, it is worth to consider, in addition to accuracy, the cost of computation hardware, and its energy consumption for training and inferring prediction models, and estimating their predictive uncertainty. Approaches such as distributed DNN, across edge-fog-cloud [26] are clearly an advantage for future sensors deployment. Nonetheless, thanks to their low-technology requirement, machine learning methods not related to the deep-learning family are still of interest.

**Random Forest (RF).** In this context, multi-timeseries AQ prediction, based on RF **for regression (RFreg)** [27], provides non-negligible results [28] that get close to the ones from DNN methods and outperform Multi-Layer Perceptron (MLP), Decision Tree, and Support Vector Machine (SVM) [19]. Methods for estimating the error prediction have been developed in the field of RF. For instance, Ref [29] (later written Wager 2014) focused on the estimation of conditional mean and variance error prediction. This work has been implemented in [30] and is widely used. Recently, Ref [31] (later written Lu 2019), inspired by [32], developed a more general and non-parametric approach that estimates conditional error prediction. As a result of their low computation requirement and their performance, both Wager 2014 and Lu 2019 are used in our study.

**AQ measurement assessment.** Scientists working with AQ sensors use methods such as [33] (later written Eaamm 2010) to assess sensors. This official approach determines whether a sensor respects DQO related to measurement error. For instance, it is used to assess low-cost sensors against robust sensors with higher accuracy and lower uncertainty, such as fixed AQ monitoring stations [2]. Instead of assessing low-cost sensors measurements, this approach can be adapted as a predictive uncertainty estimator to mutli-timeseries AQ prediction, and thus is included in our study.

### 1.4. Total Error Framework

Assessing the quality of error prediction is challenging. First, the ‘ground truth’ is usually not available. Second, scientists are aware of the effect of the error in the input while building-up prediction models [34] and try to limit their effects [35]. Third, scientists are aware of the non-perfection of their prediction models; it is now common to develop error models to be added to the prediction output [36]. Finally, prediction models represents a limited representation of a phenomenon due to our imperfect knowledge about it.

The entanglement between input, output (’ground truth’), conceptual model (e.g., regression model, Random Forest, DNN) and the different respective errors are highlighted by [37] and is adapted in Figure 1. It highlights the different errors that take part in the error prediction and how they connect to the prediction pipeline: input and output errors, structural modeling error (error of the model) and remnant errors [38,39] (i.e., error due to our imperfect knowledge of a phenomenon).

Assessing error prediction requires awareness about the different error models involved in the prediction pipeline. The Total Error framework consists of explicitly determining the model of each of these errors presented in Figure 1.

### 1.5. Objectives and Contributions

The main objectives of this paper were: (i) to introduce the field of AQ QC into the Total Error framework, (ii) to interpret each one of the three approaches (Eaamm 2010, Wager 2014 and Lu 2019) using RFreg in the Total Error framework, (iii) to identify and formulate information about each type of errors (input, output, structural modeling and remnant), (iv) to evaluate and examine the results.

### 1.6. Outline of the Paper

Section 2 describes materials and methods used in this study: Section 2.1 describes the air quality dataset from monitoring stations in Oslo, Norway used in our study, Section 2.2 describes air quality prediction using RFreg in a Total Error framework, Section 2.3 describes a diagnostic tool of the *p*-value and Section 2.4 describes the experimentation plan. Section 3 presents the results: Section 3.1 presents the metrics of the AQ prediction using RFreg, Section 3.2 presents a comparison of the three approaches (Eaamm 2010, Wager 2014 and Lu 2019) in the Total Error framework for our case study, Section 3.3 presents an analysis of the structural modeling error for RFreg and Section 3.4 presents a discussion of the results. Finally, the conclusion is presented in Section 4.

## 2. Materials and Methods

### 2.1. Air Quality Monitoring Stations in the Metropolitan Region of Oslo, Norway

The city of Oslo is a municipality of around 650,000 inhabitants located at the end of the Oslo fjord. The municipality has a surface of 480 km${}^{2}$ but the metropolitan area extends beyond its boundaries along the Oslo fjord. It is thus a population of 1.7 million inhabitants (34% of Norwegian population) living and circulating inside this area. Principal sources of urban air pollution are vehicle traffic, as $NO_{x}$ emission, and domestic wood-burning in stoves used for winter heating, as $PM_{2.5}$ and $PM_{10}$ emission. The study focuses only on $NO_{2}$. The nine AQ monitoring stations measuring the concentration of $NO_{2}$ and used in this study are presented in Figure 2 with the station metadata presented in Table 1. Eight of the monitoring stations belong to European AQ monitoring network (EOI), and one is not part of this network.

#### 2.1.1. Instrumentation of AQ Station Measuring $NO_{2}$ Concentration

AQ monitoring station reports $NO_{2}$ using an automatic $NO_{x}$ analyzer. In the region of Oslo, analyzers are model T200 from Teledyne-API [40]. The measurement method for the determination of the concentration of $NO_{2}$ and NO present in ambient air is based on the chemiluminescence measuring principle following the European reference method for measurement of Oxides of Nitrogen ([8], Annex VI). The following is a sum-up of the measurement process: in a reaction chamber, NO reacts with excess of $O_{3}$ produced inside the analyzer to form $NO_{2}$. $NO_{2}$ already present in the ambient air will not participate in the reaction. During the reaction, light with an intensity proportional to the concentration of NO is produced when electrons of the excited $NO_{2}$ molecules decay to lower energy states. The emitted light is measured by a photo multiplier tube. To measure $NO_{2}$ in ambient air, $NO_{2}$ is converted to NO in a catalytic converter before the air enters the reaction chamber and mixes with $O_{3}$. A solenoid valve switches between passing ambient air directly to the reaction chamber for measuring NO and passing the air through the converter for measuring $NO_{2}+NO$, that is $NO_{x}$. The voltage output of the photo multiplier tube is conditioned and presented as ppb NO and ppb $NO_{x}$. $NO_{2}$ is finally calculated as $NO_{x}-NO$ in ppb.

A data logger is connected to the analyzer and receives new values typically every 10 s from the instrument. The instantaneous readings are averaged to 1-h values which are stored in the data logger and transferred to the central database at NILU every hour.

#### 2.1.2. Hourly $NO_{2}$ Concentration
Dataset

The dataset of $NO_{2}$ concentration used in the study covers 4 years of hourly measurements from 2015 to 2018, with their corresponding QC label. The percentage of data coverage as described in Table 2 are close to 100% for the nine stations in between years 2015 and 2018.

As an illustration, Figure 3 represents the nine timeseries of hourly $NO_{2}$ in Oslo between the period 4 December 2015 00:00:00 and 8 December 2015 00:00:00.

#### 2.1.3. Road Work at Smestad between 2015 and 2016

In 2015–2016, the Norwegian Public Roads Administration (Statens Vegvesen) did a major upgrade of the tunnels on the motorway called ‘Ring 3’, close to the station located at Smestad. While the work was going on, the station was moved temporary from 21 May 2015 to 8 February 2017 a few meters closer to the road, to give way for a temporary footpath and bus stop. This temporarily location was 10.670066 degrees longitude and 59.932529 degrees latitude (Figure 4). As consequence, NILU collected the measurements from the two locations in two different time series to be able to separate them. It is thus expected that the concentrations at the temporary location from 21 May 2015 to 8 February 2017 to be slightly higher than at the permanent location. This event is exploited in our study.

### 2.2. Air Quality Prediction Using RFreg in a Total Error Framework

We take the case of a one-site AQ prediction based on a multi-site network using RFreg. The model predicts $NO_{2}$ concentration at one specific location and at each time step. The predictors are the other stations, located in the same city, measuring components having a connection with the $NO_{2}$ signal. Meteorological observations such as wind, temperature and other traffic-related observations are of great importance while improving the prediction of $NO_{2}$ over a city. However, for sake of simplicity, only stations measuring $NO_{2}$ are used to develop a multi-timeseries regression model. Furthermore, the regression model will not take any space-time information; no past observation will be included as predictors into the model to predict a value at time *t*. We are aware that such simplification will alter the accuracy of the prediction. Nonetheless the main goal of this study was not to provide the best prediction model but instead to introduce RFreg with the three approaches Eaamm 2010, Wager 2014 and Lu 2019 into the Total Error framework.

The purpose of this section is to highlight the different *p*-value expressions according to the error models used by Eaamm 2010, Wager 2014 and Lu 2019. All three approaches are applicable once the measurement campaign is completed. Then predicting errors can be processed either on historical data or used for real-time error prediction. All three approaches, Eaamm 2010, Wager 2014 and Lu 2019 using Random Forest, and adaptations cited in this section, are incorporated in the R-package cipred https://git.nilu.no/rqcr/cipred (accessed on 18 March 2021).

#### 2.2.1. Theoretical Approach

**Input and output errors**. Let $X={\left(X_{t}\right)}_{t=1,\cdots ,T}$ and $Y={\left(Y_{t}\right)}_{t=1,\cdots ,T}$ denote the true input timeseries (perfect signal of air pollution component at different stations) and true output timeseries (perfect signal of one specific air pollution component at one station), respectively. Let $\tilde{X}={\left({\tilde{X}}_{t}\right)}_{t=1,\cdots ,T}$ and $\tilde{Y}={\left({\tilde{Y}}_{t}\right)}_{t=1,\cdots ,T}$ denote the observed values of X and Y, respectively. We describe our input and output error as: (1)$$X=f(\tilde{X},\varepsilon_{f})$$
and
(2)$$Y=g(\tilde{Y},\varepsilon_{g})$$
where f() and g() describe the input and output error models, respectively, and $\varepsilon_{f}$ and $\varepsilon_{g}$ are their respective latent variables.

**Structural modeling error**. Models such as linear regression, non-linear regression, machine learning or any other powerful concepts, approximate a real phenomenon. Their internal algorithm is prone to artifacts that add error to the prediction. Considering a data-centric model such as Random Forest, even though inputs and outputs are perfect but are available for a limited period, it is unlikely a model will reproduce true outputs outside of the data region it has been trained for. These errors are referred to as structural modeling errors.

Let M() denote a model with a set of parameters θ that matches perfectly the relation between true input and true output. Let $\hat{Y}={\left({\hat{Y}}_{t}\right)}_{t=1,\cdots ,T}$ denote the output predicted by the model: (3)$$\hat{Y}=M\left(X,\theta \right)$$

Incorporating structural modeling errors into (Equation 3) reads:(4)$${\hat{Y}}_{t}=h\left(\tilde{M_{t}}\left(X_{t},\theta \right),\varepsilon_{h}\right)$$
where $\tilde{M_{t}}\left(\right)$ is the non-perfect model at time *t*, h() the structural error model and its latent variable $\varepsilon_{h}$. In our study, $\tilde{M_{t}}\left(X_{t},\theta \right)$ represents the supervised regression model based on the Random Forest algorithm [27] implemented in [30] where θ characterizes its set of parameters.

**Remnant error**. Our lack of knowledge about AQ phenomena and our omissions in the descriptions of our input and output errors and structural error create inevitable imperfections. These are called ‘‘remnant’’ error [38,39]. In the case of a data centric AQ modeling, this error appears because of the approximation of the impact of weather on air pollution, its coarse representation of urban activity (e.g., lack of information about work on road, etc.) and emission due to traffic activity (e.g., lack of information about traffic jam, or any accident, etc.). The remnant error reads: (5)$$Y_{t}=r\left(\hat{Y_{t}},\eta_{r},\varepsilon_{r}\right)$$
where r() is the remnant error model and its latent variables $\eta_{r}$, and $\varepsilon_{r}$. Traditional data fitting methods, such as standard least squares method, assume all observed inputs to be error free and structural and output errors to be gathered into a remnant error also called residual error being described as a Gaussian random process.

**Total Error framework**. By including Equations (Equation 1)–(Equation 3) into (Equation 5), the expression of AQ prediction in the Total Error framework reads: (6)$$g\left(\tilde{Y},\varepsilon_{g}\right)=r\left(h\left(\tilde{M_{t}}\left(f\left(\tilde{X},\varepsilon_{f}\right),\theta \right),\varepsilon_{h}\right),\eta_{r},\varepsilon_{r}\right)$$

#### 2.2.2. Approach by Eaamm 2010

**Input error.** They assume input to be error-free.

**Output error.** This error represents the measurement error of the target station monitoring AQ. The European Committee for Standardization (CEN) published a standard EN 14211:2012 [41] about ambient air monitoring instrumentation. It describes a specific method for the measurement of the concentration of $NO_{2}$ and NO by chemiluminescence, and a specific procedure for testing a candidate analyzer and thus quantifying its error. The testing requires extensive equipment and certification and is usually left to specialized testing organization like TÜV in Germany. In its report [42], TÜV presents the analyzer model T200 to get measurements $\tilde{Y_{t}}$ with an additive, heteroscedastic error proportional to the measure and following a Normal distribution. Including these information into Equation (Equation 1), AQ signal and its error read:
(7a)$$\begin{array}{cc} Y_{t} & =\tilde{Y_{t}}+\varepsilon_{g,t}, \end{array}$$
(7b)$$\begin{array}{cc} \varepsilon_{g,t} & ∼N(0,\sigma_{g,t}^{2}), \end{array}$$
(7c)$$\begin{array}{cc} \sigma_{g,t} & =\left\{\begin{array}{cc} \Phi_{g,1} & \mathrm{if}\;\;\tilde{Y_{t}}\leq \Phi_{g,3}\\ \Phi_{g,2}.\tilde{Y_{t}} & \mathrm{if}\;\;\tilde{Y_{t}}>\Phi_{g,3} \end{array}\right. \end{array}$$
where $\varepsilon_{g,t}$ is the measurement error model and $\Phi_{g,1}$, $\Phi_{g,2}$ and $\Phi_{g,3}$ its parameters.

In addition to TÜV, NILU is accredited according to ISO 17025:2017 [43] for testing analyzer by following a Standard Operating Procedures (SOPs). The estimation of the parameters is based on the results at CI95 from the TÜV report (at page 76, table 34). NILU tested the analyzer and estimated some component of the errors: the uncertainties for repeatability at zero, repeatability at the hourly limit value and linearity. The estimated standard uncertainties were 0.047, 0.287 and 2.585, respectively. Both results from TÜV and NILU about the parameters $\Phi_{g,1}$, $\Phi_{g,2}$ and $\Phi_{g,3}$ are presented in Table 3.

**Structural modeling error**. The structural modeling error is additive and follows a Normal distribution centered in zero with an unknown standard deviation. It reads:
(8a)$$\begin{array}{cc} {\hat{Y}}_{t} & =\tilde{M_{t}}\left(X_{t},\theta \right)+\varepsilon_{h,t}, \end{array}$$(8b)$$\begin{array}{cc} \varepsilon_{h,t} & ∼N(0,\sigma_{h,t}^{2}), \end{array}$$

**Remnant error**. The remnant error is highlighted by plotting a quantile–quantile chart with a reference dataset against a dataset to be tested. The remnant error represents an affine equation with a coefficient $\beta_{0}$ and a second part additive, $\varepsilon_{r}$, homoscedastic and following a Normal distribution centered in $\beta_{1}$ and with an unknown standard deviation $\sigma_{r}$. It reads:
(9a)$$\begin{array}{cc} Y_{t} & =\beta_{0}.{\hat{Y}}_{t}+\varepsilon_{r}, \end{array}$$
(9b)$$\begin{array}{cc} \varepsilon_{r} & ∼N(\beta_{1},\sigma_{r}^{2}), \end{array}$$

**Total Error framework**. By including Equations (7)–(9) into (6), the approach Eaamm 2010 in the Total Error framework reads: (10)$${\tilde{Y}}_{t}∼N\left(\beta_{0}.\tilde{M_{t}}\left(X_{t},\theta \right)+\beta_{1},\sigma_{g,t}^{2}+\sigma_{h,t}^{2}+\sigma_{r}^{2}\right)$$

Eaamm 2010 estimates first, the parameters $\beta_{0}$ and $\beta_{1}$ from Equation (9), then estimates the standard deviation gathering both $\sigma_{r}^{2}$ and $\sigma_{h,t}^{2}$, by using the following expression: (11)$$\sigma_{r}^{2}+\sigma_{h,t}^{2}=rss/\left(n-2\right)-\sigma_{g,t}^{2}+\beta_{0}+\left(\beta_{1}-1\right).{\tilde{Y}}_{t}$$
where rss is the sum of the squared of the residuals between the affine equation $\beta_{0}+\beta_{1}$. ${\tilde{Y}}_{t}$ and the prediction ${\hat{Y}}_{t}$.

Finally, the expression of the *p*-value reads: (12)$$p_{t_{eaamm2010}}=N\left({\tilde{Y}}_{t}|\beta_{0}.\tilde{M_{t}}\left(X_{t},\theta \right)+\beta_{1},\sigma_{g,t}^{2}+\sigma_{h,t}^{2}+\sigma_{r}^{2}\right)$$

#### 2.2.3. Approach by Wager 2014

**Input error.** They assume input to be error-free.

**Output error.** They assume output to be error-free.

**Structural modeling error** They assume structural modeling error being an heterocedastic error following a Normal distribution with an unknown standard deviation $\sigma_{r}$. It reads:
(13a)$$\begin{array}{cc} Y_{t} & ={\hat{Y}}_{t}+\varepsilon_{h,t}, \end{array}$$
(13b)$$\begin{array}{cc} \varepsilon_{h} & ∼N(0,\sigma_{h,t}^{2}), \end{array}$$

**Remnant errors.** They assume no remnant error.

**Total Error framework.** By including Equation (13) into (6), the expression of Wager 2014 in the Total Error framework reads: (14)$${\tilde{Y}}_{t}∼N\left(\tilde{M_{t}}\left(X_{t},\theta \right),\sigma_{r,t}^{2}\right)$$

According to [29], Wager 2014 processes the standard deviation $\sigma_{r,t}$ by estimating the variance of the bagged predictors of the RFreg.

Finally, the expression of the *p*-value reads: (15)$$p_{t_{wager2014}}=N\left({\tilde{Y}}_{t}|\tilde{M_{t}}\left(X_{t},\theta \right),\sigma_{h,t}^{2}\right)$$

#### 2.2.4. Approach by Lu 2019

**Input error.** They assume input to be error-free.

**Remnant errors.** They gather structural and output errors into a non-parametric remnant error.

**Total Error framework.** This approach provides a heteroscedastic non-parametric prediction error distribution which reads: (16)$${\tilde{Y}}_{t}∼\hat{F}\left(\tilde{M_{t}}\left(X_{t},\theta \right),v\left(X_{t}\right)\right)$$
with v(), the ensemble of the out-of-bag weights given the $\tilde{X}$ used while training RFreg model [31].

Finally, the expression of the *p*-value reads: (17)$$p_{t_{lu2019}}=\hat{F}\left(\tilde{Y_{t}}|\tilde{M_{t}}\left(X_{t},\theta \right),v\left(X_{t}\right)\right)$$

### 2.3. Predictive qq-Plot: Diagnostic to *p*-Values

**Performance of the error prediction.** Instead of a metric such as mean-square-error and all the relatives, error predictions are validated using the *p*-value of the observations $\tilde{Y}$. The predictive qq-plot (pqq-plot) provides a simple and informative summary of the performance of error prediction. It has been used as a verification tool for hydrological and meteorological prediction and simulation [38,39,44,45]. We summarize here the concept: If the observation ${\tilde{Y}}_{t}$ is a realization of the predictive distribution described by its cumulative distribution function (cdf) ${\hat{F}}_{t}\left(\right)$, every *p*-value ${\hat{F}}_{t}\left({\tilde{Y}}_{t}\right)$ is a realization of a uniform distribution on [0,1]. The pqq-plot compares thus the empirical cdf of the sample of *p*-values ${\hat{F}}_{t}\left({\tilde{Y}}_{t}\right)$ for every time step with the cdf of a uniform distribution. An illustration of the pqq-plot is adapted from [45] and presented in Figure 5; it can be interpreted as follows:–Case 1: If all points fall on the 1:1 line, the predicted distribution agrees perfectly with the observations.–Case 2: the window of the predicted error is overestimated.–Case 3: the window of the predicted error is underestimated.–Case 4: the prediction model systematically under-predict the observed data.–Case 5: the prediction model systematically over-predict the observed data.–Case 6: When an observed *p*-value is 1.0 or 0.0, the corresponding observed data lies outside the predicted range, implying that the error prediction is significantly underestimated.

**Precision of the error prediction**, also called sharpness, is a term used while being in case 1. It is a qualitative description of how narrow the error prediction distribution is while including the ’ground truth’. A higher precision describing a narrow error prediction is helpful for taking decision.

### 2.4. Experimentation Plan

#### 2.4.1. Comparison of the Three Approaches

The first experimentation focus on comparing models of error prediction according to the three approaches Eaamm 20110, Wager 2014 and Lu 2019 for each station presented in Table 1 but the one located at Smestad. While one station is the target, the other stations represent the predictors. The model does not use any space-time information; no past observation will be included as predictors into the model to predict a value at time *t*. The intern parameters of the Random Forest algorithm are kept as the default ones from the Ranger library [30] (number of trees: 500, number of variables to possibly split at in each node: 2, minimal node size: 5). The purpose of the study is not to improve the RFreg prediction. The second experiment focus on highlighting the effect of changing the location of a station. The target station is the one located at Smestad, the other stations represent the predictors.

For the diagnosis of both the RFreg prediction and the error prediction, we focus on valid observed hourly $NO_{2}$ data only, as determined by quality control experts from NILU. The full observed dataset is split in two equal and continuous parts: the first period is used for the training and the testing of the model; the second period is used for the validation. The first period is randomly split in two parts with a ratio 80/20 for the training and the testing steps. Given road works at Smestad between 2015 and 2016, we decide to choose the training and testing phase for years 2017–2018 and the validation phase for years 2015–2016.

We apply regular metrics to quantify the general accuracy of the predicted output from RFreg: Root Mean Square Error (rmse), Mean Absolute Error (mae), Bias and coefficient of determination ($R^{2}$).

To compare the error predictions of the three approaches, we plot their output on a pqq-plot.

#### 2.4.2. Quantifying the Structural Modeling Error of RFreg with Synthetic Datasets

Quantifying each part of the Total Error framework is useful to give insights to QC AQ experts on how much they can trust the error prediction and which part of the errors might be failing in producing relevant information.

In our specific case, input and output errors got their expression quantified by a certified organization (TÜV). In addition, Eaamm 2010 showed remnant error to follow an affine equation. We focus this second experiment in providing more information about the structural modeling error of RFreg.

The work of [46,47] highlights the presence of structural modeling error in RFreg. In order to illustrate the structural modeling error in RFreg in a more general case, we create two synthetic datasets representing a “truth”: (i) dataset A representing the study of [46,47] with one input characterized by an uniform probability density function (pdf) and a linear expression connecting input and output, (ii) dataset B with 10 inputs, each of them following an exponential pdf with parameters from 0.01 to 0.055 and a non-linear expression connecting inputs and output. In all our experiments, input and output are scaled and assumed perfect, i.e., without any noise. The parameters of the synthetic datasets are described in Table 4. The RFreg model is trained on a subset of the “truth”. Two types of subset are created: (i) type 1 with values chosen uniformly in the middle of the dataset, (ii) type 2 with values chosen randomly. We avoid any remnant error by including all the inputs in the training and the prediction phases. The parameters of the training dataset are described in Table 5. To reproduce the results from [46,47], we process a first experiment by using dataset A and train our RFreg on training dataset subset type 1. We then predict using the input of the whole dataset A. We process a second experiment by using dataset B and train our RFreg on training dataset subset type 2. We then predict with the inputs from dataset B not used in training dataset. We repeat the second experiment four times to get an overview of structural modeling error RFreg with datasets having similar characteristics.

From the three approaches used in this study, Wager 2014 provides an expression of the structural modeling error independent from the other errors. In addition, the construction of the error prediction from Lu 2019 enables to catch the structural modeling error in the case where only this type of error is present. We thus implement both Wager 2014 and Lu 2019 in our second experiment to look at how these approaches catch the error signal. The work of Eaamm 2010 determined indirectly structural modeling error from input and remnant errors and empirical residual. It will not be included in this part of the study.

## 3. Results

### 3.1. Metrics of AQ Prediction Based on RFreg

Metrics about the RFreg prediction is presented in Table 6. For the testing phase, RFreg provides moderate prediction for most of the stations with a $R^{2}$ around 0.75, a bias close to zero and low rmse. Only station 11 sees its RFreg prediction getting a low $R^{2}$ at 0.57. For the validation phase, all the stations but station 11 gets a $R^{2}$ around 0.75 with a light decrease in comparison to the testing phase. The bias is slightly larger than during the testing phase and are in a range between −5.2 and 4.7. As expected, station 504 sees its bias decreasing to −11.95.

### 3.2. Comparison of the Three Approaches

An illustration of the timeseries at station 7, including the observations, the RFreg prediction and the different error predictions at CI95 during the validation period is presented in Figure 6. The error prediction looks wider with Eaamm 2010 than with Lu 2019 and thinner with Wager 2014. This impression is confirmed in the pqq-plot Figure 7 where all the stations are present. During the testing phase and according to Figure 5, Eaamm 2010 overestimates the window of the predicted error, while Wager 2014 underestimates the window of the predicted error. Lu 2019 agrees with the observations. During the validation phase, Lu 2019 provides a better error prediction in comparison to Eaamm 2010 and Wager 2014 but sees the uniform distribution of its *p*-values being slightly distorted.

An illustration of the timeseries at station 11, between 4 December 2016 and 8 December 2016 and characterized by an RFreg prediction model with a low $R^{2}$ during the validation period, is presented in Figure 8. This low metric does not alter the error prediction both for the training and the validation phases, as we see in Figure 9.

An illustration of the timeseries at station 504, between 4 December 2015 and 8 December 2015 and characterized by the change of location during the validation period, is presented in Figure 8. The error model Lu2019 agrees with the observations in the testing phase, but systematically under-predict the observed data in the validation phase, as seen on Figure 10.

### 3.3. Structural Modeling of RFreg

The result of our experiment with the synthetic dataset A and subset type 1 is illustrated in Figure 11. It reproduces the results of [46,47]. We see two phases in this figure. The first one represents the RFreg model correctly predicting the output. The second one shows RFreg reaching its limit at the border of the sampling dataset used for training the model. Indeed, the prediction describes two constants determined by the output at the limit of the sampling. Like RFreg, its corresponding error shows two phases: a plateau with a value around zero inside the sampling area and two vertical asymptotes.

The illustrations of the errors for our second experiment with synthetic dataset B and subset type 2 are presented from Figure 12, Figure 13, Figure 14 and Figure 15. The errors do not describe a distinct two-phase behavior anymore. Instead, the plateau gets noisier the further it gets away from the dense area of the output used for the training. The error then “waves” as a transition phase until getting to the asymptote. Both the error predictions of Wager 2014 and Lu 2019 get a wider error CI95 window the further it gets away from the dense area of output. Contrarily to the Wager 2014 approach, which follows the waving signal with a symmetric window, Lu 2019 reaches an upper and lower constant in an asymmetric way. The pqq-plot of our second experiment with synthetic dataset B and subset type 2, both from Wager 2014 and Lu 2019, during the prediction period, are presented in Figure 16. Lu 2019 provides a better error prediction in comparison to Wager 2014.

### 3.4. Discussion

**Accuracy of the RFreg prediction**. In our case study, the results indicate that a simple one-site AQ prediction based on a multi-site network using RFreg provides moderate metrics. Former studies [19,28] showed that adding meteorological observation as input will increase accuracy, as for example getting $R^{2}$ closer from 0.9.

**Performance of the error prediction**. The results indicate that among the three approaches used in this study, Lu 2019 provides an error prediction with a predicted distribution that best agrees with AQ monitoring stations. This is reflected by a density of its *p*-values following the pattern of Case 1 in Figure 5.

**Precision of the error prediction**. In addition to the performance, the precision of the error prediction, also described as its “sharpness”, is of great importance for QC experts. A higher precision describing a narrow error prediction is helpful for taking decision. For instance, the error prediction based on Lu 2019 for station 11 shows a good performance. Due to an RFreg model getting poorer metrics, the error prediction describes a wider error CI95 window in comparison to the error prediction on other stations. Increasing the precision of error prediction for this station will require an investigation into new features affecting the AQ signal and incorporating them in the RFreg model. To a larger extent, ensuring an error prediction with a high precision requires efforts on getting accurate input, output and prediction model, and limiting our lack of knowledge about a “true” phenomenon.

**Quantifying each error type in the Total Error framework**. Quantifying each type of error involved in the error prediction is useful for assessing the error prediction model and further improving it, in terms of performance and precision.

The expression of input and output errors results in experiments in a highly controlled environment on AQ monitoring stations. The expression of remnant error results from the study of Eaamm 2010. By its construction, Wager 2014 is the approach that provides an expression for quantifying the structural modeling error of RFreg. By testing Wager 2014 and comparing it with Lu 2019 in a specific case where only structural modeling error occurs, it is possible to highlight potential changes for improving the description of structural modeling error. A better performance and precision might be reached by (i) enabling a model error following a non-parametric or a non-Gaussian and asymmetric distribution, (ii) getting the error model following an upper and a lower constant after a prediction threshold. In addition, both Wager 2014 and Lu 2019 get difficulties in predicting larger errors related to high values of prediction. Further work will focus on improving the structural modeling error model of RFreg.

Finally, input is considered error-free in the three approaches. Further work will be required to include it into the Total Error framework.

**Towards a spatial error prediction**. As shown in the study with the case of the station in Smestad, a temporarily short shift of a station location alters considerably the ability of our error prediction to provide good results. Further work will focus on reducing the location-dependency of our prediction model. First, reducing the bias of the prediction model will be done through the use of Land Use regression (LUR) model that enables to catch small-scale spatial variation of air pollutants within urban areas [48]. Indeed, spatial features, such as topography, proximity to infrastructure as well as the land use in a buffer area near to air quality monitoring stations, have proven to improve the performance of deep learning-based air pollution predictions [49]. DNN LUR, such as [50] provide an approach of geo-context in an unsupervised way that outperforms other approaches such as RF. For these reasons, spatial error prediction will require the use of either Bayesian DNN or Deep Ensemble.

## 4. Conclusions

Instead of a flag valid/non-valid usually proposed in the quality control (QC) processes of air quality (AQ), we proposed a method that predicts the *p*-value of each observation as a value between 0 and 1. We based our error predictions on three approaches: Eaamm 2010, Wager 2014 and Lu 2019.

Total Error framework enables to differentiate the different errors: input, output, structural modeling and remnant. We thus theoretically described a one-site AQ prediction based on a multi-site network using Random Forest for regression in a Total Error framework.

We implemented the methodology with a dataset of hourly $NO_{2}$ measured by a network of monitoring stations located in Oslo, Norway, and implemented the error predictions for the three approaches. The results indicate that a simple RFreg prediction model provides moderate metrics for fixed stations. According to the diagnostic based on pqq-plot and among the three approaches used in this study, Lu 2019 provides better error predictions. Our RFreg prediction model sees limitations while temporarily shifting a station location. Further work will be required on reducing the location-dependency of our prediction model.

Quantifying each type of error involved in the error prediction is useful for staying critical about the error prediction models and further improving them in terms of performance and precision. We presented the structural modeling error of RFreg with synthetic datasets aiming at reproducing existing results and proposing improvement in the expression of its error model. Further work will focus on improving the structural modeling error model of RFreg. Finally, input is considered error-free in the three approaches. Further work will be required to include it into the Total Error framework.

## Figures and Tables

**Figure 1 sensors-21-02160-f001:**
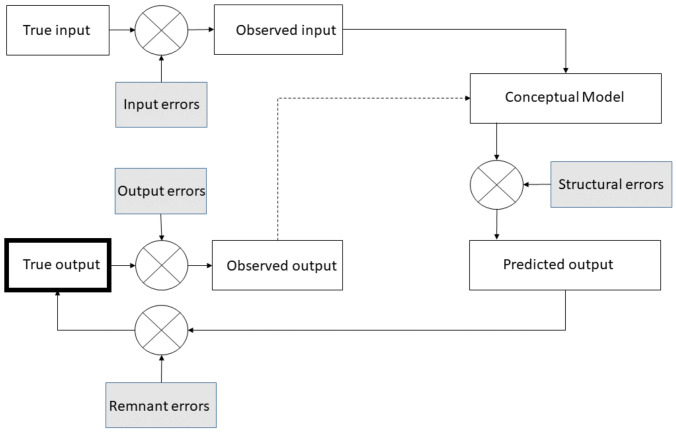
Pipeline in between input, output, a conceptual model and its prediction. Dashed arrow corresponds to the need of output for the training of the model.

**Figure 2 sensors-21-02160-f002:**
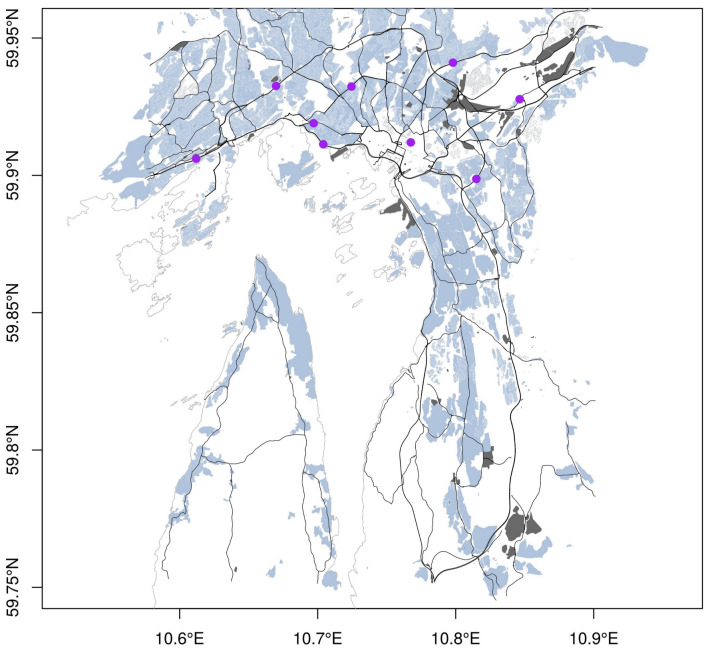
Map of Oslo with its residential (blue) and industrial (dark gray) areas, its roads (primary, secondary, motorway and trunk) and its air quality (AQ) station network measuring $NO_{2}$ (purple circle). GIS data come from Open-Street Map.

**Figure 3 sensors-21-02160-f003:**
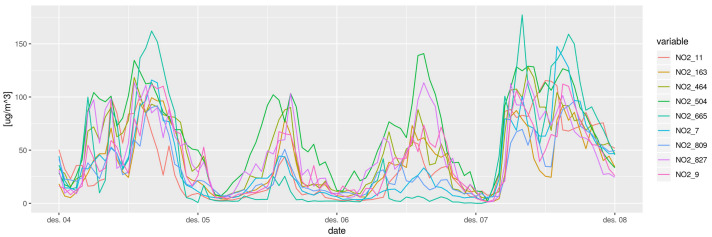
$NO_{2}$ observation timeseries between 4 December 2015 00:00:00 and 8 December 2015 00:00:00 of the nine monitoring stations in the municipality of Oslo.

**Figure 4 sensors-21-02160-f004:**
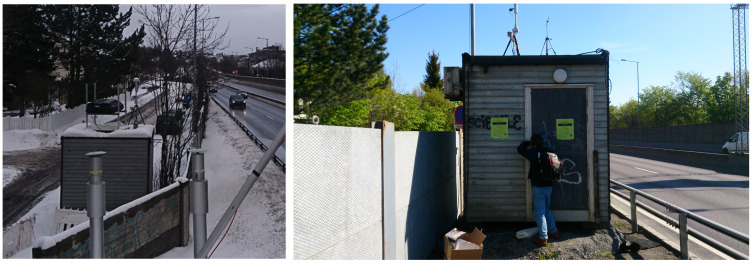
Permanent location of station 504 at Smestad (**left**). Temporary location of station 504 from 21 May 2015 to 8 February 2017 (**right**).

**Figure 5 sensors-21-02160-f005:**
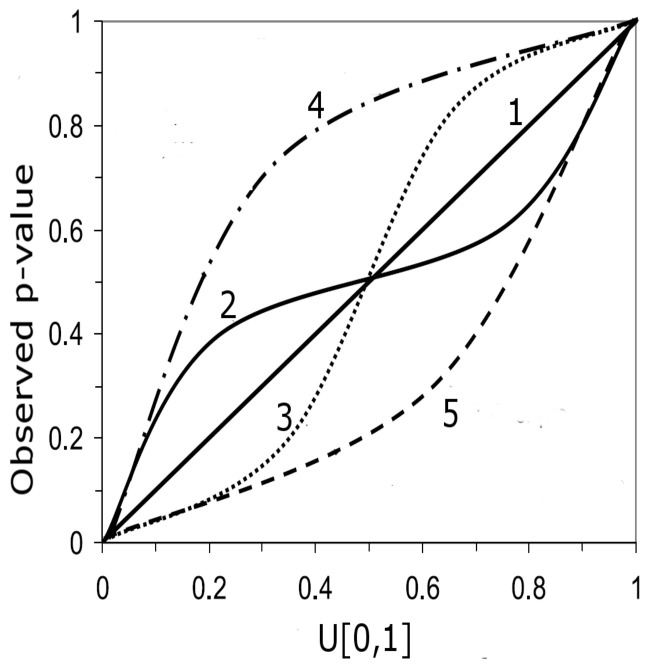
Interpretation of the predictive qq (pqq)-plot adapted from [45].

**Figure 6 sensors-21-02160-f006:**
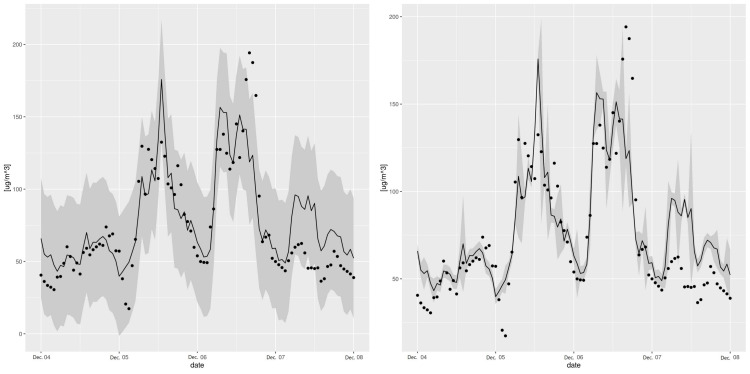
Sample of observation (dots) timeseries with the RFreg prediction (line), the error prediction at CI95 (gray area) following Eaamm 2010 at station 7 between 4 December 2016 and 8 December 2016 (**upper left**), following Wager 2014 at station 7 between 4 December 2016 and 8 December 2016 (**upper right**), following Lu 2019 at station 7 between 4 December 2016 and 8 December 2016 (**lower**).

**Figure 7 sensors-21-02160-f007:**
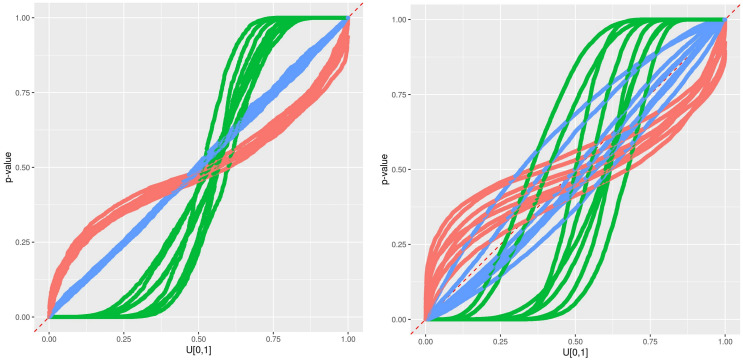
Predictive qq-plot for the training/testing period 2017–2018 (**left**). Predictive qq-plot for the validation period 2015–2016 (**right**). The different methods are colored in red for Eaamm 2010, green for Wager 2014 and blue for Lu 2019. Each curve corresponds to one station.

**Figure 8 sensors-21-02160-f008:**
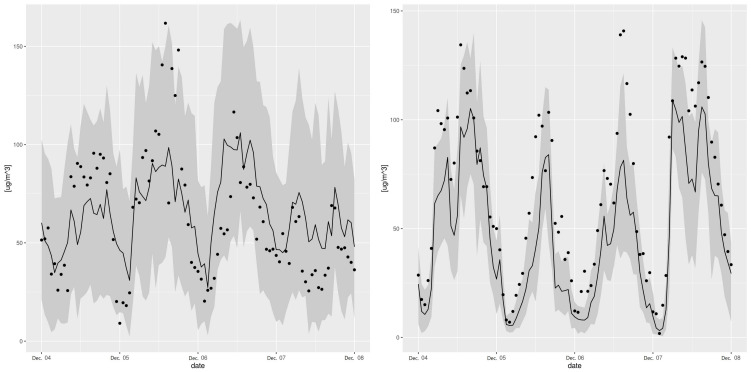
Sample of observations (dots) timeseries with the RFreg prediction (line), the error prediction at CI95 (gray area) following Lu 2019 at station 11 between 4 December 2016 and 8 December 2016 (**left**), following Lu 2019 at station 504, one year earlier, between 4 December 2015 and 8 December 2015 (**right**).

**Figure 9 sensors-21-02160-f009:**
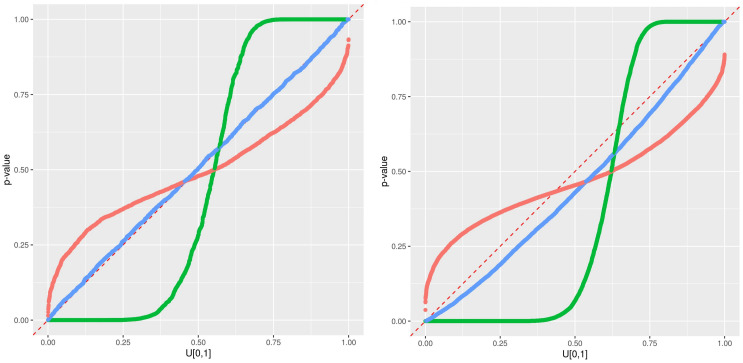
Results at Station 11. Predictive qq-plot for the training/testing period 2017–2018 (**left**). Predictive qq-plot for the validation period 2015–2016 (**right**). The different methods are colored in red for Eaamm 2010, green for Wager 2014 and blue for Lu 2019.

**Figure 10 sensors-21-02160-f010:**
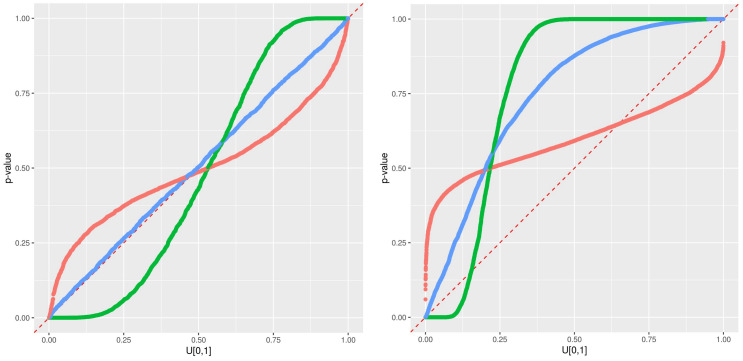
Results at Station 504. Predictive qq-plot for the training/testing period 2017–2018 (**left**). Predictive qq-plot for the validation period 2015–2016 (**right**). The different methods are colored in red for Eaamm 2010, green for Wager 2014 and blue for Lu 2019.

**Figure 11 sensors-21-02160-f011:**
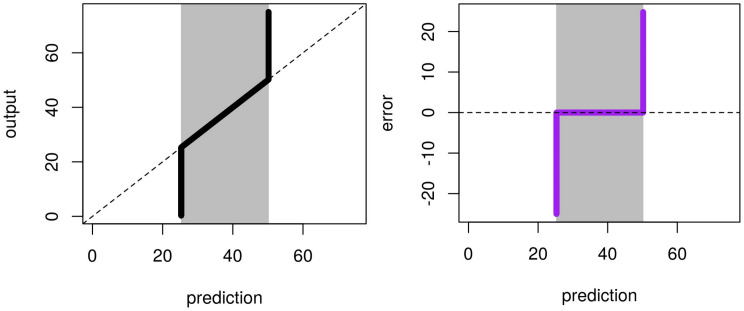
Illustration of the artifact of RFreg on structural modeling error with dataset A and subset type 1. Artfact on the prediction is shown on the **left**. Artifact on the error is presented on the **right**. Vertical gray lines represent output subset used for the training phase.

**Figure 12 sensors-21-02160-f012:**
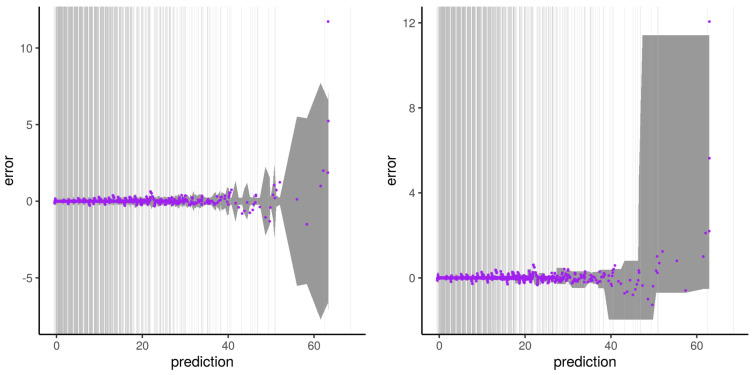
Illustration of the artifact of RFreg on structural modeling error with dataset B and subset type 2. Artifact on the error (purple) and Wager 2014 error prediction CI95 (dark gray) is shown on the **left**. Artifact on the error (purple) and Lu 2019 error prediction CI95 (dark gray) is shown on the **right**. Vertical gray lines represent output subset used for the training phase.

**Figure 13 sensors-21-02160-f013:**
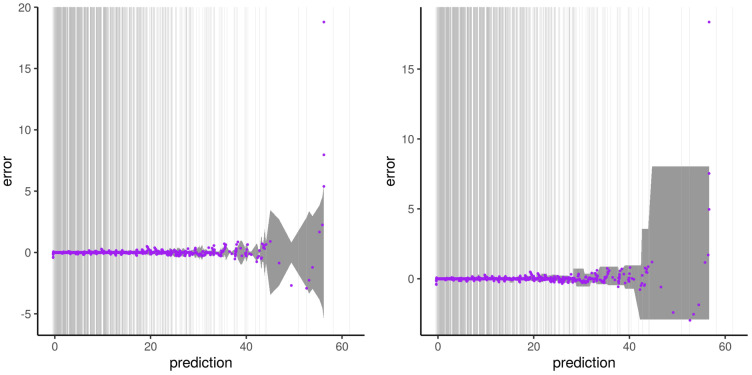
Same as Figure 12 for the second run of the experiment.

**Figure 14 sensors-21-02160-f014:**
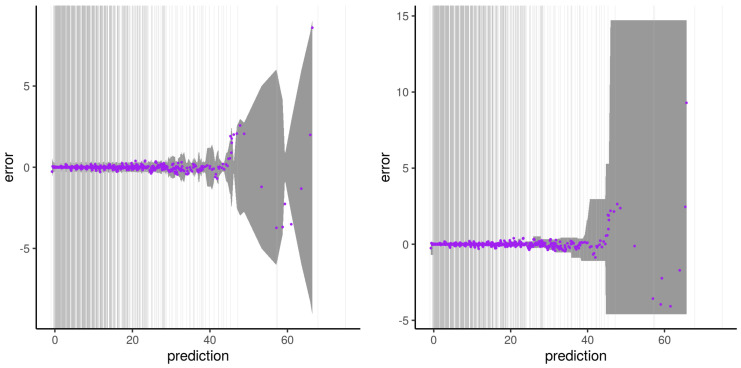
Same as Figure 12 for the third run of the experiment.

**Figure 15 sensors-21-02160-f015:**
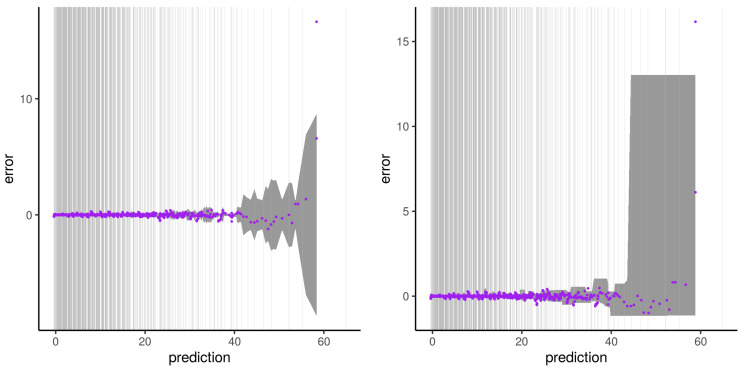
Same as Figure 12 for the fourth run of the experiment.

**Figure 16 sensors-21-02160-f016:**
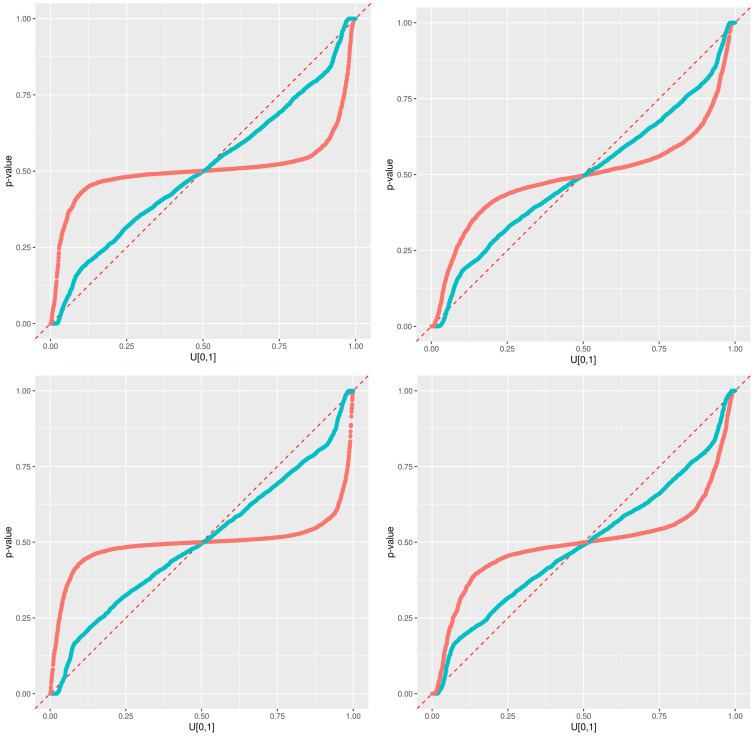
Predictive qq-plot for the prediction period with dataset B and subset type 2. The different methods are colored in red for Wager 2014 and blue for Lu 2019.

**Table 1 sensors-21-02160-t001:** Metadata of the nine monitoring stations measuring $NO_{2}$ located in the metropolitan region of Oslo.

Name	ID	Municipality	Coordinates (Lon/Lat)	Area Class	Station Type	EOI
Alnabru	7	Oslo	(10.84633, 59.92773)	suburb	near-road	NO0057A
Bygdøy Alle	464	Oslo	(10.69707, 59.91898)	urban	near-road	NO0083A
Eilif Dues vei	827	Bærum	(10.61195, 59.90608)	suburb	near-road	NO0099A
Hjortnes	665	Oslo	(10.70407, 59.91132)	urban	near-road	NO0093A
Kirkeveien	9	Oslo	(10.72447, 59.93233)	urban	near-road	NO0011A
Manglerud	11	Oslo	(10.81495, 59.89869)	suburb	near-road	NO0071A
Rv 4, Aker sykehus	163	Oslo	(10.79803, 59.94103)	suburb	near-road	NO0101A
Smestad	504	Oslo	(10.66984, 59.93255)	suburb	near-road	NO0095A
Åkebergveien	809	Oslo	(10.76743, 59.912)	urban	–	–

**Table 2 sensors-21-02160-t002:** Metadata of the nine monitoring stations measuring $NO_{2}$ located in the municipality of Oslo.

ID	Component	2015–2018	
		Coverage (%)	Valid (%)
7	$NO_{2}$	99	93
464	$NO_{2}$	99	90
827	$NO_{2}$	99	99
665	$NO_{2}$	99	98
9	$NO_{2}$	92	92
11	$NO_{2}$	99	99
163	$NO_{2}$	97	96
504	$NO_{2}$	98	98
809	$NO_{2}$	99	90

**Table 3 sensors-21-02160-t003:** Parameters of the measurement error expression. $\Phi_{g,1}$ and $\Phi_{g,3}$ are expressed in ${g/m}^{3}$, $\Phi_{g,2}$ is a percentage. Values from NILU come from an internal technical report. Parameters $\Phi_{g,1}$ and $\Phi_{g,2}$ are divided by 1.96 in order to get a generalization on the whole Normal distribution and not only at the confidence interval 95-percentile (CI95). Parameter $\Phi_{g,3}$ is a threshold and does not require any division by 1.96.

Entity	$\Phi_{g,1}$	$\Phi_{g,2}$	$\Phi_{g,3}$
TÜV	0	4.35/1.96	0
NILU	5.64/1.96	5/1.96	112.8

**Table 4 sensors-21-02160-t004:** Characteristics of “truth” datasets with expression between input and output, linear: $y=1/40x_{0}$, non-linear: $y=x_{0}+x_{1}+x_{2}+log\left(x_{3}\right)-x_{4}+x_{5}/x_{6}+x_{7}+log\left(x_{8}\right)/x_{9}$, with $x_{9}=1$ if $x_{9}\;\mathrm{mod}\;2=0$, and $x_{9}=50$ otherwise.

ID	Length	Input	Pdf		y=f(x)
			Distribution	Parameters	
A	2991	1	uniform	–	linear
B	2991	10	exponential	[0.01:0.055]	non-linear

**Table 5 sensors-21-02160-t005:** Characteristics of training datasets subset, built-up from “true” datasets.

Type	Length	Index Range	Choice of the Indexes
1	1000	[1001:2000]	non-random, no replacement
2	1000	[1:2991]	random, no replacement

**Table 6 sensors-21-02160-t006:** Metrics of the prediction of the nine monitoring stations measuring $NO_{2}$ for the testing phase and the validation phase.

	Testing			Validation		
**ID**	**rmse**	**Bias**	$R^{2}$	**rmse**	**Bias**	$R^{2}$
7	12.92	0.10	0.76	15.49	4.70	0.76
464	10.13	−0.13	0.79	13.82	−5.21	0.80
827	11.33	0.42	0.74	15.37	0.79	0.70
665	16.26	−0.53	0.71	3.34	−2.04	0.67
9	7.34	0.13	0.85	12.35	−3.20	0.80
11	18.22	0.81	0.57	22.01	2.71	0.54
163	10.31	−0.29	0.76	13.91	0.17	0.72
809	8.16	0.33	0.79	9.98	1.18	0.80
504	9.66	−0.10	0.83	19.94	−11.95	0.77

## Data Availability

Source code of our library cipred is available at https://git.nilu.no/rqcr/cipred (accessed on 18 March 2021). Docker with cipred and all library dependencies to reproduce the experiments from the paper is located at https://hub.docker.com/r/jmll/cipred_paper (accessed on 18 March 2021). Code for running the experiments are located at https://git.nilu.no/rqcr/cipred/-/tree/master/inst/paper (accessed on 18 March 2021). Data used in this paper is located at https://git.nilu.no/RextData/data.luftkval.oslo10 (accessed on 18 March 2021).

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
