# Peer review of "Error Prediction of Air Quality at Monitoring Stations Using Random Forest in a Total Error Framework"

_sensors, 2021, doi:10.3390/s21062160_

Round 1

Reviewer 1 Report

The paper is not innovative enough. In terms of references, it does not include recent papers on air quality control. Moreover, the author focuses on regression prediction, but the results given are not predictions of the air-related index.
Reviewers are willing to discuss the following issues with the author:
1) Why use three methods? This way of writing is not a good way of writing a paper. Regarding the comparison of the performance of these three methods, the method proposed in a paper is not innovative enough. It is recommended to use only the best method among them. 2) How is the random forest-based regression method proposed by the author achieved? The description in the text is not clear. 3) The regression method based on the random forest is not the latest research result of air quality prediction. The author should mention the latest research method in related methods, such as the method based on deep learning and other machine learning method. Some important references must be added: “Shishegaran, A., Saeedi, M., Kumar, A. and Ghiasinejad, H., 2020. Prediction of air quality in Tehran by developing the nonlinear ensemble model. Journal of Cleaner Production, 259, p.120825.”, " Harandizadeh, H. and Armaghani, D.J., 2021. Prediction of air-overpressure induced by blasting using an ANFIS-PNN model optimized by GA. Applied Soft Computing, 99, p.106904.", “Jin, X.B., Yu, X.H., Su, T.L., Yang, D.N., Bai, Y.T., Kong, J.L. and Wang, L., 2021. Distributed Deep Fusion Predictor for a Multi-Sensor System Based on Causality Entropy. Entropy, 23(2), p.219.” the author should discuss, why not use deep learning, neural networks, and other methods that can model more complex nonlinear models, but use regression methods based on random forests. 3) The framework of Figure 1 needs further explanation. In the following chapters, there is no detailed explanation of the framework and what are the advantages of the framework. 4) The author should explain how to evaluate the performance of the prediction results and how the reference data is obtained. 5) How to determine the parameters in Equation 7? 6) The value of R^2 in the results given in Table 6 is not greater than 0.9, indicating that the reliability of this result is not strong. 7) The captions of Figure 9 and Figure 10 are exactly the same and should be distinguished. 8) What is the difference between the performance of the prediction error and the precision of the prediction error in Section 3.4?

Reviewer 2 Report

The authors have studied the error from using Random Forest for regression (RFreg) based on data measured air quality monitoring sites to predict at other station. The error prediction is quantified at each observation by a p-value using 3 different approaches to determine the error in the Total Error framework . The method is interesting and can be valuable for other type of data prediction besides air quality. 

The authors should expand the discussion regarding the application of the method such as in spatial prediction or interpolation of observed data from fixed locations to unknown location.

There are a few minor errors or obscure wordings that should be clarified

(1) Line 3-4 : References for these 3 approaches in the Total Error framework should be mentioned here. There is no reference for Eaamm2010 in the reference section.     

(2) Line 9: RFreg - explain it as Random Forest for regression (RFreg)

(3) Line 14; "to stay critic" should be "to assess"

(4) Line 28-33: The use of listed items (i) (ii)...(vi) in the sentence is difficult to follow. Listed them out in a list or delete/replace these items i ,ii... vi with (i), (ii)...

(5) Line 43-47: Use parentheses around these listed items i.e (i) (ii)... instead of i-, -ii in the sentence.

(6) Line 51: "massively" should be "extensively"

(7) Line 53: "amount" should be "number"

(8) Line 181: "Mt(-,..) should be "Mt(Xt,..)"

(9) Table 3: "intern technical report" should be "internal technical report"

(10) Line 368, 397: "staying critic" should be "assessing"

Reviewer 3 Report

The authors theoretically described a one-site AQ prediction based on a multi-site network using Random Forest for regression in a Total Error framework. They compared their error predictions with those of three other papers. No doubt, error prediction is an important tool for testing the efficacy of a prediction methodology. This is an interesting and a very well written paper.

1. Why did you use the random forest algorithm?
2. I suggest that besides random forest algorithm, you select e.g. two more AI methods (e.g. deep learning and support vector machine, or other) and compare the results obtained and decide which procedure is the most suitable, which error prediction is best?
3. Do you think that, of the three procedures you have to examine, the most appropriate method for making an error prediction for urban air pollutants other than NO2 (e.g. ozone, PM2.5, etc.) is the finally selected one, or are methods used for this purpose pollutant-specific?
4. It would be nice if you cited similar papers in the literature and compared your results with those of the cited papers in the Discussion section.

Round 2

Reviewer 1 Report

The text in Figures 7, 10, and 16 is too small.

Author Response

Point 1) The text in Figures 7, 10, and 16 is too small.

Answer 1) We did modify Figure 7, Figure 9, Figure 10, and Figure 16
according to your comment. Explanations about the colours and their corre-
sponding approach are given in each caption.

Reviewer 3 Report

Dear Authros,

I accept the revised version of the manuscript as it is.

Reviewer

Author Response

Point 1) I accept the revised version of the manuscript as it is.
Answer 1) Then, no new modification got implemented.